# Physical Changes of Preschool Children during COVID-19 School Closures in Fujian, China

**DOI:** 10.3390/ijerph192013699

**Published:** 2022-10-21

**Authors:** Guobo Li, Le Yang, Xingyan Xu, Mingjun Chen, Yingying Cai, Yeying Wen, Xiaoxu Xie, Xinyue Lu, Suping Luo, Shaowei Lin, Huangyuan Li, Siying Wu

**Affiliations:** 1Department of Child Healthcare Centre, Fujian Maternity and Child Health Hospital, College of Clinical Medicine for Obstetrics & Gynecology and Pediatrics, Fujian Medical University, Fuzhou 350001, China; 2School of Public Health, Fujian Medical University, Fuzhou 350122, China; 3Department of Developmental and Behavioral Pediatrics, Fujian Children’s Hospital, Fuzhou 350014, China

**Keywords:** weight, BMI, COVID-19, preschoolers, public health prevention

## Abstract

The COVID-19 pandemic may constitute an “obesogenic lifestyle” that results in exacerbating childhood obesity. However, studies investigating regional sociodemographic factors including different age groups or sexes in children with obesity are lacking. We aimed to clarify the high obesity prevalence populations of preschool children to provide a regional basis for children’s health policy during the COVID-19 school closures. From May to September 2019, a total of 29,518 preschool children were included in a large sample, multicenter cross-sectional study to explore physical status in Fujian Province by stratified cluster random sampling. In October 2019 and October 2020, we also conducted a cross-sectional study exploring physical development including changes in height, weight, and BMI of 1688 preschool children in Fuzhou before and after the COVID-19 school closures. Student’ s *t*-test, Mann–Whitney U test, or chi-square test was used to assess differences in physical development and overweight and obesity rates among preschool children before and after school closures. For regional factors, the weight of urban preschool children of all ages became higher after the outbreak (*p* (age 3–4) = 0.009; *p* (age 4–5) < 0.001; *p* (age 5–6) = 0.002). For sex factors, overweight and obesity in boys had a greater prevalence than in girls before and after the outbreak. In four age groups, overweight and obesity rates in the 5-year-old group (15.5% and 9.9%) were higher than before (11.4% and 6.0%). The weight and BMI of 4- to 5-year-old children also increased faster than before (*p* < 0.001). The COVID-19 pandemic has promoted the epidemic of childhood obesity. Living in urban/coastal (economically developed) areas, boys, and aged 4–6 years old may be a susceptible population to obesity development after the outbreak.

## 1. Introduction

China, with its fast economic growth, has witnessed a rapid increase in childhood obesity rates, which causes a major public health problem affecting children’s physical and mental health [1]. Compared with developed countries, the rate of overweight and obesity among children in China was relatively low, but the total population of obese children has ranked first in the world [2]. In 2002 and 2008, Fujian Province carried out two nutritional surveys on children under 7 years old, and the detection rates of childhood obesity were 4.9% and 6.4%, respectively, both higher than the national level in the same period [3]. In the epidemiological survey of childhood obesity in nine cities of China, the obesity rate of children under 7 years old in Fuzhou, the capital city of Fujian Province, was higher than that of three first-tier cities, Beijing, Shanghai, and Guangzhou (5.5% vs. 4.7%, 4.4%, 3.2%) [4], indicating that the prevention and control situation of childhood obesity in Fujian Province is quite severe.

The world is currently facing serious challenges from the infectious disease of COVID-19, improving measures and activities for prevention and control [5]. The COVID-19 pandemic may constitute an “obesogenic lifestyle” that leaves us vulnerable to an unprecedented obesogenic environment, resulting in exacerbating childhood obesity [6,7,8,9,10]. Canadian scholars have found that the COVID-19 pandemic may reduce the health impact of weight management programs for children and adolescents, especially for boys, leading to increased weight and BMI [11]. A multicenter study in China found that the trend in childhood obesity was intensifying in the context of the COVID-19 pandemic, and that altered weight-related behaviors such as less daily physical activity, reduced sleep duration, and longer screen time due to lockdown measures may have largely contributed to this trend [12]. The main influencing factors of obesity include genetic, individual, social, and environmental factors [13]. Among these, the latter two changeable factors may be affected by changing lifestyles during the COVID-19 epidemic. Public health prevention and corresponding policies, such as school closures and subsequent home isolation, lead to dietary and exercise changes that may negatively impact children’s physical and mental health [14,15]. The previous study has suggested that the COVID-19 pandemic might double the time children spend outside school, exposing them to a different environment than usual [16]. Multiple studies have demonstrated that children were getting less physical activity on non-school days which may contribute to excessive weight gain or even obesity [14,17]. If left unchecked, it can lead to a major challenge that threatens long-term human health and economies, which exerts a far-reaching influence over COVID-19 illness [18].

With a higher probability of staying obese in adulthood than children without obesity, children with obesity are more likely to experience non-communicable diseases in the future [19]. Moreover, in children with overweight and obesity, the age of weight growing fastest varied from 2 to 6 years old. Therefore, preschool age is a critical period for adjusting the energy homeostasis of the human body to prevent childhood obesity. Making targeted interventions in early periods in life is of great significance for actively preventing childhood obesity progression, which in turn prevents chronic diseases, obesity, and related conditions in adulthood. A study explored the changes in weight and height among Chinese preschool children during COVID-19 school closures and concluded that the prevalence of childhood obesity increased [20], but it did not investigate childhood obesity according to regional sociodemographic factors including different age groups or sexes.

Here, we explored the prevalence of physical status of preschool children before the COVID-19 outbreak in Fujian, China and measured the physical development of preschool children before and after the outbreak. The purpose of this study was to clarify the prevention and control focus of childhood obesity. This large sample, multicenter analysis may provide a regional basis for formulating effective prevention and control strategies for childhood obesity.

## 2. Methods

### 2.1. Study Design and Participants

We conducted a cross-sectional study of overweight and obesity in preschool children in Fujian, China using a stratified cluster random sampling method from May to September 2019. First, the cities to be investigated were selected from 9 cities in Fujian Province (Fuzhou, Putian, Ningde, Quanzhou, Zhangzhou, Longyan, Sanming, and Nanping, a total of 8 cities, except Xiamen). Secondly, 1 to 4 districts and counties were randomly selected from each selected city. Finally, in each district and county under investigation, with kindergarten as the unit, 3 to 5 kindergartens were randomly selected from the cluster. All children aged 3 to 6 in the kindergarten were selected as the research objects of this project to explore the overweight and obesity rates of public preschool children.

Meanwhile, we used cluster random sampling to select 2 urban areas and 2 suburban areas of Fuzhou, and from each selected 2 kindergartens. All children aged 3 to 6 in the kindergartens were selected as the research objects of this project and height and weight measurements of preschoolers at the start of the pandemic (October 2020). Before the outbreak of the epidemic in early January 2020, kindergartens of Fuzhou had started winter vacation and classes were closed successively. Therefore, we collected the children’s baseline data in October 2019 as the data for the “normalization period”. With the easing of the epidemic, kindergartens resumed school in September 2020. Thus, we collected the data in October 2020 as the data for “the start of school after the epidemic”. Matching the physical examination results of the same child in 2019 and 2020, the growth value of each growth indicator between the two years was calculated. In addition, the physical examination data of the same kindergarten in the same period from 2018 to 2019 were collected as data on the growth of children during the normalization period (non-epidemic period) and were compared with the data of the epidemic period to explore the physical status of preschool children before and after school closing due to the epidemic. Figure 1 shows the flow diagram of the participants of the PART I province-wide survey (before the COVID-19 outbreak) and PART II of physical development of preschool children (before and after the outbreak). See Appendix B and Appendix C for sample size calculations and investigation areas.

The inclusion and exclusion criteria were as follows. Inclusion criteria: (1) voluntarily participation in the study and signing the informed consent; (2) subjects cooperate to complete the physical examination; (3) no secondary obesity caused by congenital genetic diseases such as congenital heart disease, digestive system diseases, metabolic diseases, neurological and endocrine diseases; (4) no obvious physical deformity. Exclusion criteria: (1) refusal to participate in the study; (2) failure to cooperate or complete the health examination; (3) secondary obesity caused by congenital genetic diseases such as congenital heart disease, metabolic diseases, and neurological and endocrine diseases; (4) obvious developmental abnormalities, such as severe scoliosis, lameness, obvious O-leg and X-leg, etc.

Children 3 to 6 years old who were enrolled in a participating school were eligible for inclusion. Pupils with a physical disability or secondary or syndromic causes of obesity were excluded. The study was approved by the Ethical Committee of Fujian Medical University (approval No.2019-117).

### 2.2. Measurements

Participants were weighed in kilograms in light clothing and bare feet. The weight was measured with a standard scale with an accuracy of 0.1 kg, and the height was measured with a rangefinder with an accuracy of 0.1 cm. Both height and weight were measured twice and recorded to ensure reliability. The determination of childhood obesity or overweight includes the reference commonly used for physical examination standards in China [21], the indicator of “weight value for height” to evaluate the physical development level of children. According to the “weight value for height” obtained by body testing, children who are ≥110% of the standard weight are screened out. After the diagnosis of the child health medical examiner, the final diagnosis is made after the pathological or secondary obesity is excluded: ≥110% of the standard weight and <120% of the standard weight = children with overweight, ≥120% of the standard weight = children with obesity. From these data, we calculated body mass index (BMI), as well as changes in BMI, weight, and height. Demographic information includes the school district, gender, and age.

### 2.3. Statistical Analysis

Continuous variables were analyzed as mean and standard deviation or the median and interquartile range while categorical variables were analyzed as frequency and percentage. A Student’ s *t*-test, Mann–Whitney U test, or chi-square test was used to assess differences in physical development and overweight and obesity rates among preschool children before and after school closures. Subgroup analyses included age strata, school locale (suburban vs. urban), and sex. Statistics were analyzed by the chi-square test for trends in age strata. The Bonferroni method was used for pairwise comparisons. All analyses were prespecified and performed using SPSS 26.0. Significance was assessed at the 5% level.

## 3. Results

A total of 30,485 children were included in the PART I province-wide survey (before the COVID-19 outbreak). Of these, 632 participants did not complete the entire physical examination. Through the verification of the data, valid data were obtained from 29,518 participants, 16,009 male students (54.2%), and 13,509 female students (45.8%) who participated in the survey, see Appendix A. In PART II physical development of preschool children before and after the outbreak in October 2020, 1688 children with the requirements were included. There were 931 male students (54.7%) and 757 female students (44.8%). In October 2019, we also collected the growth data of 4540 children in the same kindergarten during the pre-epidemic period from 2018 to 2019, including 2503 boys (55.1%) and 2037 girls (44.9%), see Appendix A.

### 3.1. Physical Status of Preschool Children before the COVID-19 Outbreak in Fujian, China

The detection rate of overweight among preschool children in Fujian Province was 10.2%, among which the detection rate of overweight in boys was 11.5%, which was higher than that of girls (8.8%, *p* < 0.001). The detection rate of obesity was 6.6%, of which it was 8.0% in boys, higher than the 4.8% in girls (*p* < 0.001). The detection rates of overweight among preschool children in coastal areas (Fuzhou, Quanzhou, Putian, Zhangzhou, and Ningde) and mountainous areas (Sanming, Longyan, and Nanping) of Fujian Province were 10.6% and 9.1%, respectively (*p* < 0.001). The detection rate of obesity in coastal areas was 6.8%, higher than the 5.7% in mountainous areas (*p* = 0.001). Overweight and obesity rates in boys and girls in coastal areas were higher than those in girls and boys in mountainous areas. The rates in boys were higher than those in girls in the same area, see Figure 2. In the 6-year-old group, the overweight rate in urban areas was higher than that in suburbs, see Appendix A.

Chi-square tests for trend showed that the overweight and obesity rates of preschool children in Fujian Province presented an increasing trend with the increase in age (*p* < 0.001). The pairwise comparison demonstrated that the obesity rate of the 6-year-old group was not statistically different from that of the 5-year-old group, nor was the obesity rate of the 3-year-old group compared with the 4-year-old group, whereas the detection rate of obesity in the 5- to 6-year-old group was higher than that in the 3- to 4-year-old group (*p* < 0.001). Between the ages of 3 and 6, the detection rate of overweight in boys was higher than that in girls. In the 4-year-old group, 5-year-old group, and 6-year-old group, the detection rate of obesity in boys was higher than that in girls, see Appendix A.

### 3.2. Physical Development of Preschool Children of Different Age Groups before and after the Outbreak

In the baseline data of the 4-year-old group, the 5-year-old group, and the 6-year-old group, there was no statistical difference in the detection rate of overweight and obesity in 2018 and 2019, see Appendix A. After the outbreak of the COVID-19 epidemic, the detection rates of overweight and obesity in the four groups of preschool children (aged 3 to 6 years) were statistically different from those in the corresponding age groups before the outbreak. Pairwise comparison noted that the detection rate of overweight in the 3-year-old group was 7.2%, which was lower than that before the outbreak (13.7%), while the difference in the detection rate of obesity between the two periods was not statistically significant. The detection rate of overweight in the 4-year-old group was 18.6%, which was higher than that before the outbreak (12.6%), while the difference in the detection rate of obesity between the two periods was not statistically significant. The detection rate of overweight and obesity in the 5-year-old group after the outbreak was 15.5% and 9.9%, which was higher than before the outbreak (11.4% and 6.0%). The detection rate of obesity in the 6-year-old group after the outbreak was 11.2%, which was higher than that before the outbreak (6.0%), while the difference in overweight rates between the two periods was not statistically significant. The health policies related to the COVID-19 have caused dramatic changes in the lifestyles of preschool children. At the same time, the government has also proposed relatively complete emergency family education plans for children’s home lifestyles, including specific recommendations on a scientific diet and reasonable exercise, see Figure 3.

After the outbreak, the growth values of height, weight, and BMI in the 3- to 4-year-old group were higher than those before the outbreak. The increases in weight and BMI in the 4- to 5-year-old group were higher than those before the outbreak. The 5- to 6-year-old group had a higher growth value for height than normal, while BMI growth was lower than normal. The results for boys were consistent with those in the general population. For girls, the growth value of height in the 5- to 6-year-old group was higher than that in the normal period, while the growth values of weight and BMI were lower than those in the normal period (*p* < 0.05), see Table 1.

### 3.3. Physical Development of Different Sexes after the Outbreak

After the outbreak, the height growth of boys in the 3- to 4-year-old group was lower than that of girls, and there was no statistically significant difference in weight and BMI growth between boys and girls. In the 4- to 5-year-old group, there was no statistical difference in height, weight, and BMI increase between boys and girls. However, in the 5- to 6-year-old group, the increase in weight of boys was higher than that of girls, and there was no significant difference in the increase in height and BMI between boys and girls, see Table 2.

### 3.4. Physical Development of Different Regions after the Outbreak

After the outbreak, the growth values of height and weight of urban children in the 3- to 4-year-old group were higher than those of suburban children (*p* < 0.05). The weight and BMI growth values of urban children in the 4- to 5-year-old and 5- to 6-year-old groups were higher than those of suburban children, and the height growth of urban children was lower than that of suburban children (*p* < 0.05), see Table 3.

## 4. Discussion

To curb the spread of the COVID-19 epidemic, the government continues to issue public health policies such as home isolation and reducing unnecessary travel. The COVID-19 school closures will cause negative implications for the physical development of preschool children. Our study investigated the growth of preschool children before and after the outbreak to assist in the prevention of childhood obesity.

### 4.1. Regional Factors

Our studies demonstrate that overweight and obesity rates among preschool children in mountainous areas are lower than in economically developed coastal areas. The development of childhood obesity might be associated with regional economic levels. One study suggests that in developing countries, the risk of childhood overweight/obesity increased in families of a better economic situation [22]. Our province-wide survey before the outbreak showed higher overweight rates in urban areas in the 6-year-old group. After the outbreak, the weight of urban preschool children of all ages grew more than that of suburban children, while the height growth of 4- to 6-year-old urban children was lower than that of suburban children. Due to limited space or opportunities for physical exercise, children living in urban areas are more likely to gain weight [8], which has a negative effect on physical development. Therefore, children in urban and coastal areas need far more attention when developing a control strategy for childhood obesity.

### 4.2. Sex Factors

Boys are the focus of childhood obesity prevention and control before and after the outbreak. Between urban and suburban areas, coastal areas and mountainous areas, overweight and obesity in boys were more prevalent than in girls. This may relate to traditional Chinese sexual culture and societal cognitive biases about body size [23]. A study demonstrates that Chinese parents preferred a heavier ideal body image for their boys which leads to an epidemic of boys’ obesity [24]. However, parents are more likely to be influenced by the social notion that girls should remain slim, and focus on diet control helping obesity control. A cohort study of 445 Chinese children aged 7–12 years showed that BMI was more likely to increase in boys during the 5-month COVID-19-related isolation period [25]. A simulation study of the effects of COVID-19 on childhood obesity and BMI in the United States also showed that it affected boys more than girls [26]. Before the pandemic, boys tended to spend more time on physical activity than girls, especially moderate and intense physical activity [27]. However, during COVID-19 family confinement and school closures, boys experienced greater decreases in physical activity and greater increases in sedentary behavior [6,28]. Meanwhile, with the gender differences in variation in food intake, girls ate more vegetables and fruits, while boys increased their intake of processed meat during the pandemic [29]. All of these reasons may lead to faster weight and BMI growth in boys during the COVID-19 pandemic than in girls. Thus, gender differences should also be taken into account in health care policies during the epidemic.

### 4.3. Age Groups

We found that as age increased, a trend of increasing obesity rates in preschool children was observed. One longitudinal study reported that it was difficult for children who already had obesity at an early age to transform to overweight or normal weight during subsequent growth [19]. Our study findings add support to the necessity of early childhood obesity prevention and control.

Our research shows that the weight, height, and BMI of the 3-year-old children after COVID-19 school closures were lower than before the outbreak. This phenomenon may be due to the fewer opportunities for going out shopping during the school closure period, leading to difficulties in nutrient intake to meet the growing needs of children. Deficiency in micro- and macronutrients causes slow growth and even nutritional diseases [30,31]. In addition, home isolation reduced the outdoor activity time of preschool children, which lowers the sunshine time, leading to a lack of vitamin D and affecting their height growth [7,32].

After the outbreak, the weight and BMI of 4- and 5-year-old children increased faster than before the outbreak, indicating that these two age stages may be the focus of obesity prevention during the epidemic. The survey before the outbreak showed that the rates of overweight and obesity in the 3- and 4-year-old groups were similar, and increased rapidly in the 5- and 6-year-old stages. After the outbreak, children with overweight and obesity in the 4-year-old group had already developed rapidly relative to the 3-year-old group, showing a different distribution from the condition before the outbreak. This may be due to the dramatic change in children’s lifestyles during COVID-19 school closures.

### 4.4. Lifestyle Change by Public Health Policy

Children may be exposed to more unhealthy (high-sugar, high-salt, and high-fat) diets during school closures [33]. Due to the poor access to fresh food by the inconvenience of going out, many households choose to buy and store processed foods with a long shelf life to reduce going out [16]. These highly processed foods often contain large amounts of saturated fat, sugar, and salt [34]. A study performed in Italy found that consumption of potato chips, red meat, and sugar-sweetened beverages had increased significantly during home isolation [27]. Moreover, some parents lost their jobs during the pandemic, leading to financial hardship and changing diet choices. Such socioeconomic changes may negatively affect children’s eating habits [18].

A lack of structured physical activity in children during school closures causes an increase in sedentary behavior and weight [35,36]. Recently, quite a few studies have demonstrated that increased screen time may also cause profound negative health effects, leading to a higher risk of children with overweight and obese [37,38]. Online learning and a limited range of activities at home increase children’s daily screen time [14]. A study has shown that there changes in food intake with an increase in screen exposure may cause childhood obesity [39]. However, not all families have worse eating habits during the epidemic. Instead, some families see this quarantine period as a perfect opportunity to cook home-cooked food and expose children to more fruits and vegetables to change poor dietary preferences [40,41]. Therefore, it is necessary to provide corresponding advice on the behavior of parents during home isolation for childhood obesity.

A study by Brazendale et al. [42] suggested that behaviors that lead to obesity, such as long sedentary time, increased screen time, poor diet, and irregular sleep patterns, will improve if the child lives regularly. Health care and education systems require both a policy and health care response for childhood obesity. Parents need to be taught how to choose food scientifically. Moreover, sufficient physical activity is required while maintaining social distancing.

Relevant national agencies need to screen for, monitor, prevent, treat, and manage childhood obesity for individuals and groups [43,44], and introduce special intervention and treatment guidelines to curb the health harm caused by overweight and obesity to children and reduce the public health burden on children. In addition, nutrition and health knowledge education among children, adolescents, and their parents should be strengthened, to help them establish healthy eating habits and concepts. Effective interventions can increase knowledge of health practices and improve self-perception in obese children and adolescents, which may develop into sustained behavior change. Overweight and obesity are largely preventable. Supporting policies, the school, and the community are the key for parents and children to make a choice [45,46]. The government should provide supportive policy options to reduce obesity in schools. These may include the development of policy options focused on nutrition/physical education and parental involvement in nutrition and physical activities [47]. The government should allow parents, teachers, and community health workers to participate in developing or revising childhood obesity control strategies, to bolster comprehensiveness and strength of the district’s wellness policy language as well as accountability for these changes [48]. The strategy for capitalizing on opportunities to increase availability of schoolyard equipment should be promoted to enhance students’ physical activity. The health management department should work with after school program decision makers to ensure that the schedule includes 50% or more of the time being allocated to vigorous physical activity [48]. Health education for school students and their parents includes the following measures: (1) the use of posters in the classroom to teach children about healthy food and to encourage them to be physically active; (2) the provision of information skills such as recipe books or picture cards that increase children’s awareness and knowledge of healthy eating and physical activity; (3) enabling the teachers have good command theories of behavioral change to assist in maximizing children’s knowledge of activities during class time; (4) providing parents with information about healthy eating and physical activity through regular newsletters; (5) encouraging parents to be part of decision-making related to healthy school nutrition and the physical activity program.

During the epidemic, children cannot go out, so we should pay attention to reasonable exercise while doing good diet management. Children can keep fit at home and choose some indoor sports they like. Parents can let their children do some housework within their power, such as mopping the floor, washing dishes, and wiping the table. Parents should use this time to increase parent–child interaction, parent–child games, etc. To improve physical fitness (PF) in obese children during COVID-19 restrictions, exercise specialists are starting to provide physical training through remote exercise. Online supervised training programs can effectively promote physical activity (PA), improve PF, and reduce BMI Z-scores in obese children [49].

Therefore, preventing and managing childhood obesity should be a priority for individuals, families, and communities during the pandemic.

### 4.5. Strengths and Limitations

To the best of our knowledge, few studies analyzing weight and height changes in preschool children during the COVID-19-related school closure period exist. A major strength of our study was the rigorous research design, which was used to explore focal populations who need childhood obesity prevention. We also discuss the impact of public health policy on lifestyle changes in preschool children after the outbreak and on physical changes. However, our study should be considered in the context of important limitations. In this study, there were differences in several baseline characteristics in the 6-year-old group, that could have contributed to differences in results. Thus, we used the growth of each variable in analysis to reduce errors. A limitation of observational studies, including the current study, is lacking measure factors to explain associations, thus we only make assumptions based on the status quo. Further studies in larger populations and using exposure assessments including physical activity levels and dietary intake are needed to confirm the preliminary results.

## 5. Conclusions

Overall, the COVID-19 pandemic could fundamentally change society’ s way of life to promote the obesity epidemic. Children’s weight and BMI increased faster after the outbreak in the 4- to 5-year-old children, which suggests the need for focus on obesity prevention during the epidemic. After the outbreak, living in urban areas, boys, and aged 4–6 years old may be a susceptible population for obesity development, with a higher detection rate of obesity or overweight. These findings may inform both national and regional established public health policies. Effective public health policies are urgently needed to reduce the risk of childhood obesity during the pandemic. Nutrition and health guidance should be strengthened for key groups in the prevention and control of childhood obesity, so as to reduce the impact of long-term home living on their physical and mental health during the epidemic and the possible adverse health consequences such as obesity and chronic diseases. Ongoing monitoring of growth in children and adolescents should be performed in the next phase to examine the current progression of obesity and its adverse outcomes over time.

## Figures and Tables

**Figure 1 ijerph-19-13699-f001:**
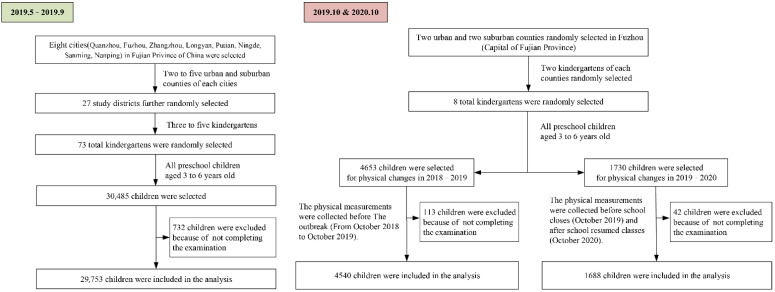
Children’s recruitment flow chart.

**Figure 2 ijerph-19-13699-f002:**
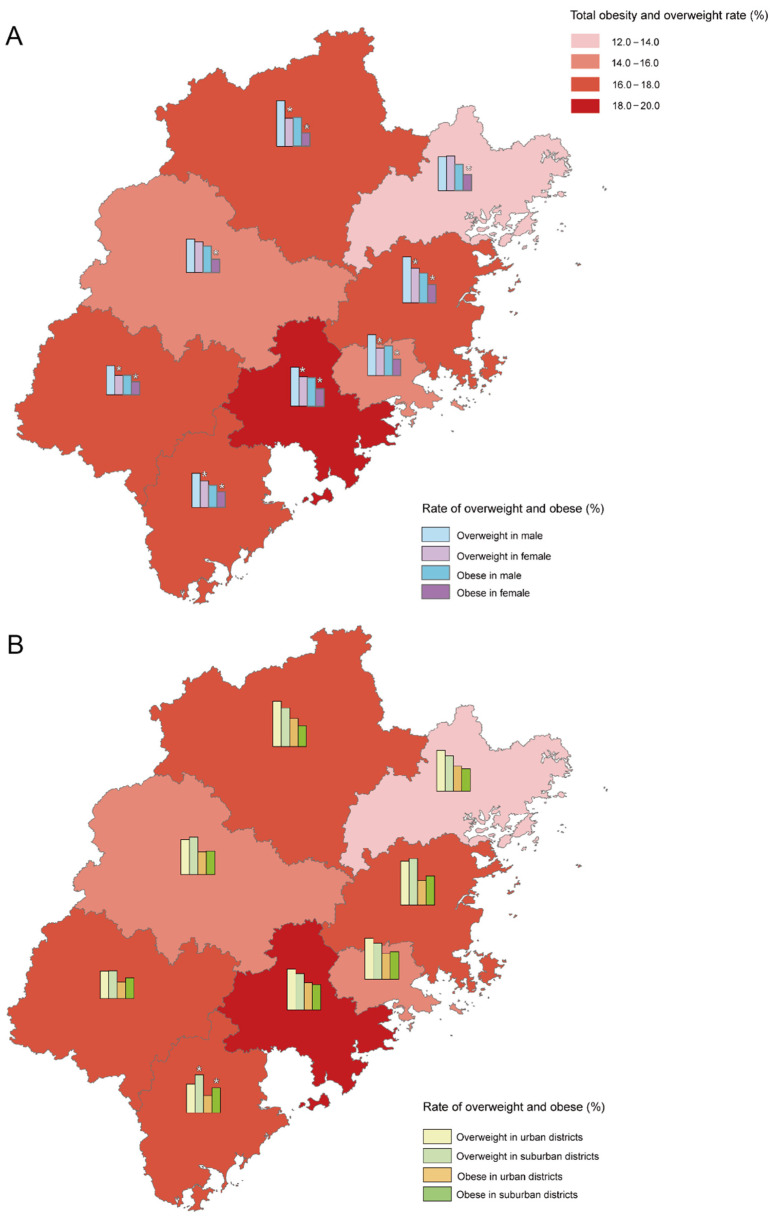
Overweight and obesity rates of preschool children in Fujian, China. (**A**) Comparison of overweight and obesity rates of different sexes. (**B**) Comparison of overweight and obesity rates in urban and suburban areas. * *p* < 0.05.

**Figure 3 ijerph-19-13699-f003:**
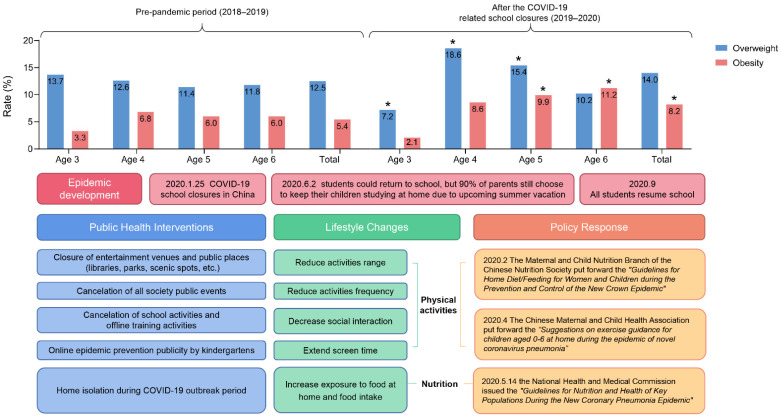
Changes in lifestyle and physical status of preschool children before and after the outbreak. *: Comparison of overweight and obesity rates before and after the outbreak (*p* < 0.05).

**Table 1 ijerph-19-13699-t001:** Characteristics of physical development of participating children during the prepandemic period and after the outbreak (*n* = 1688).

Variable	Age 3–4		t/Z	*p*	Age 4–5		t/Z	*p*	Age 5–6		t/Z	*p*
	Prepandemic	After the Outbreak			Prepandemic	After the Outbreak			Prepandemic	After the Outbreak		
Total	1307	435			1143	706			669	215		
Height, cm	7.76 ± 1.79	9.31 ± 1.55	−17.344	<0.001	6.85 ± 1.88	7.00 ± 3.79	−0.931	0.352	6.27 ± 2.02	7.69 ± 7.40	−4.483	<0.001
Weight, kg	2.71 ± 1.17	3.56 ± 1.61	−10.086	<0.001	2.43 ± 1.40	2.80 ± 1.41	−5.485	<0.001	2.66 ± 1.37	2.67 ± 1.33	−0.178	0.859
BMI	0.08	0.24	−4.195	<0.001	0.56	0.15	−3.589	<0.001	0.29	0.04	−4.787	<0.001
	(−0.38–0.59)	(−0.21–0.89)			(−0.53–0.64)	(−0.28–0.68)			(−0.16–0.81)	(−0.45–0.67)		
Girl	615	180			484	308			294	88		
Height, cm	7.87 ± 1.76	9.53 ± 1.73 *	−11.110	<0.001	6.92 ± 1.83	6.91 ± 1.40	0.074	0.941	6.20 ± 1.95	7.12 ± 1.35	−8.076	<0.001
Weight, kg	2.68 ± 1.14	3.48 ± 1.45	−6.772	<0.001	2.34 ± 1.25	2.69 ± 1.24	−3.888	<0.001	2.64 ± 1.32	2.35 ± 0.92 *	0.233	0.021
BMI	0.01 (−0.35–0.50)	0.21 (−0.26–0.68)	−2.536	0.011	0.03 (−0.54–0.51)	0.11 (−0.30–0.64)	−2.279	0.023	0.35 (−0.14–0.89)	0.10 (−0.42–0.32)	−4.972	<0.001
Boy	692	255			658	398			375	127		
Height, cm	7.66 ± 1.81	9.16 ± 1.40	−13.466	<0.001	6.81 ± 1.91	7.07 ± 4.89	−1.187	0.236	6.32 ± 2.07	8.08 ± 9.56	−3.353	0.001
Weight, kg	2.73 ± 1.18	3.61 ± 1.72	−7.494	<0.001	2.50 ± 1.50	2.88 ± 1.52	−4.003	<0.001	2.67 ± 1.40	2.89 ± 1.53	−1.511	0.116
BMI	0.08 (−0.38–0.59)	0.24 (−0.21–0.88)	−3.264	0.001	0.07 (−0.53–0.63)	0.15 (−0.28–0.68)	−2.819	0.005	0.25 (−0.19–0.80)	0.04 (−0.46–0.68)	2.109	0.035

*: There are statistical differences in height and weight development of different sexes (*p* < 0.05).

**Table 2 ijerph-19-13699-t002:** Characteristics of physical development of participating children of different sexes after the outbreak (*n* = 1688).

Variable	Age 3–4		t/Z	*p*	Age 4–5		t/Z	*p*	Age 5–6		t/Z	*p*
	Girl (*n* = 180)	Boy (*n* = 255)			Girl (*n* = 308)	Boy (*n* = 398)			Girl (*n* = 88)	Boy (*n* = 127)		
Height, cm	9.53 ± 1.73	9.16 ± 1.40	2.429	0.016	6.91 ± 1.40	7.06 ± 4.89	−0.616	0.538	7.12 ± 1.35	8.08 ± 9.56	−1.047	0.295
Weight, kg	3.48 ± 1.45	3.61 ± 1.72	−0.846	0.398	2.69 ± 1.23	2.88 ± 1.52	−1.787	0.074	2.35 ± 0.92	2.89 ± 1.52	−3.255	0.001
BMI	0.21 (−0.26–0.68)	0.24 (−0.21–0.88)	−1.158	0.247	0.11 (−0.30–0.64)	0.15 (−0.28–0.68)	−1.132	0.258	−0.10 (−0.42–0.32)	0.04 (−0.45–0.67)	−1.888	0.059

**Table 3 ijerph-19-13699-t003:** Characteristics of physical development of participating children in urban and suburban areas after the outbreak (*n* = 1688).

Variable	Age 3–4		t/Z	*p*	Age 4–5		t/Z	*p*	Age 5–6		t/Z	*p*
	Urban (*n* = 184)	Suburban (*n* = 251)			Urban (*n* = 555)	Suburban (*n* = 151)			Urban (*n* = 104)	Suburban (*n* = 111)		
Height, cm	9.53 ± 1.70	9.15 ± 1.42	2.538	0.011	6.77 ± 1.21	7.84 ± 7.81	−3.204	0.001	6.91 ± 1.25	8.42 ± 10.19	−4.167	<0.001
Weight, kg	3.81 ± 1.86	3.37 ± 1.38	2.624	0.009	2.91 ± 1.44	2.38 ± 1.18	4.584	<0.001	2.96 ± 1.43	2.41 ± 1.19	3.063	0.002
BMI	0.23	0.24	−1.194	0.232	0.19	−0.12	4.64	<0.001	0.18	−0.19	2.819	0.005
	(−0.22–0.93)	(−0.25–0.69)			(−0.23–0.70)	(−0.54–0.34)			(−0.35–0.71)	(−0.49–0.33)		

## Data Availability

The datasets presented in this article cannot be made openly available due to legal and ethical reasons. The authors welcome requests for collaboration. Requests to access the datasets should be directed to the corresponding author or lgb0703@163.com. Requests may be subject to ethics approval and/or participant consent.

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
