# Peer review of "Physical Changes of Preschool Children during COVID-19 School Closures in Fujian, China"

_ijerph, 2022, doi:10.3390/ijerph192013699_

Round 1
Reviewer 1 Report
The authors try to investigate physical changes in pre-school children during school closures due to COVID-19, analysing regional socio-demographic factors, including different age groups or sexes, in children with obesity, to provide strategies for the prevention and control of childhood obesity on a regional basis. The manuscript is generally well written and easy to read; a slight spell-check is required.
Abstract
line 18-20: "To clarify the high obesity prevalence populations of preschool children for providing a regional basis for children health policy during the COVID-19 school closures”. " I think this is the aim of your study, so you should state it (i.e. we aimed to..)
keywords usually should be different from that used in the main title.
Introduction
The literature on the subject is sufficiently well summarised, but it can be improved.
Methods
The methods section is sufficiently well described.
Validity of the findings
The results and discussion section are quite clear and organised. The parameters considered are well presented.
In section 4.2 Sex factors, the results observed in section 3.3 should be reported and discussed more clearly
Author Response
Dear Professor,
Thank you for your comments concerning our manuscript. We have made a point-by-point response to the your comments. Please see the attachment.
We appreciate your warm work earnestly and hope that the correction will meet with approval.
Once again, Special thanks to you for your comments and suggestions.

Reviewer 2 Report
First of all, thank you for the opportunity to review this article. Second, I would like to congratulate the authors of the article who have done a great job in conducting this investigation.
In my review, I would like to make some comments on each part of the article separately.
Introduction
The introduction is short, the authors concentrate on the main idea of the study. The authors' desire to focus more on what is related to the research context by justifying the research problem is understandable. On the other hand, this study is relevant not only in the studied country, but also in a wider context. Therefore, the introduction (and at the same time the justification of the research problem) would be enriched if the authors paid more attention to other studies that investigated the impact of COVID on preschool children overweight and obesity.
Methods
The organization of the study is clearly described. The supplementary material, which justifies the study sample and sampling, is particularly significant.
One interesting aspect. Why was Fujian province chosen for the study?
Results
The authors of the article provide a lot of valuable data (some of them are presented in the supplementary files). I would like to draw attention to several aspects. It is very difficult (almost impossible) to read the information next to the maps in the Figure 2. Just pay attention to it. Not quite clear Table S3. Is there really a gender comparison here? Also the meanings of the letters a,b,c. For example, the meaning of a was not so clear for me (I could just guess what it means, but not sure). So, a more detailed explanation would be good (it just suggestion to think about).
Discussion
It is good that the discussion section is structured. Data on overweight and obesity among preschool children are discussed from different perspectives: regional, gender, age.
The authors of the article also discuss about public health policies. This is an important part of the discussion. The authors present certain recommendations for prevention of childhood obesity. However, these recommendations are more general in nature. In other words, I miss recommendations more related to a situation like COVID-19. If it is possible to add, it would be interesting and valuable.
Conclusions
Conclusions are clear.
Author Response

(The authors gave the same response as above.)

Reviewer 3 Report
· Line 26: It is unclear what does it mean by Page?
· The statistical analysis used in the study should be clearly stated in abstract.
· The introduction is not complete or is not clear enough to read. In all paragraphs there is information that does not “set the stage” directly for the research aim. The section would benefit from a review of more literature regarding the obesity and PA patterns and changes during the COVID-19 lockdown, not only in China, but also in other countries. I do believe many cross-sectional and longitudinal studies are conducted in this area, particularly among school children. There is also clear lack of novelty or the importance of conducting the study. What this study adds? Why in China? Why not in other Asian countries?
· The method does not reflect very well the research aim. It is not completely clear whether the authors originally aimed to conduct a cross-sectional study or longitudinal study. This is quite confusing. PA and BMI are assessed at two time points.
· Line 73: “an epidemiological survey”???- Please clarify.
· It would be useful to have an overview about preschools. Were they public or private? Were they classified as low, middle or high SES? How were the schools from each site selected?
· Line 103-104: Please expand on the inclusion and exclusion criteria. How authors deal with missing values?
· Figure 1 has poor resolution. Please replace with a better image.
· Line 123-131: Did the authors perform the normality test to see if the data are normal distribution?
· Line 144-152- meaning unclear- no need to include details on circRNAs, please delete.
· Please be consistent throughout- change male/female to boys/girls.
· The discussion is very week, short and devoted of recent references to literature. Please discuss your results in light of more recent studies (please refer to my comment in introduction).
· Line 284-311: It is good to include lifestyle change by public health policy. However, this section should be expanded. It would be benefit to provide environmental strategies/policy recommendations to reduce childhood obesity. Authors should look at short and long term policies with anticipated outcomes for the public health problem, with some reflection on whole of government and/or intersectoral recommendations. I would suggest referring these articles (e.g., Children (Basel). 2018 Jan 29;5(2):18.doi 10.3390/children5020018; Int J Environ Res Public Health. 2020, 17(22):8405).
· Line 327-333: The conclusion needs to cohesively summarize the importance of the authors’ work as well as the implications for future studies.
· English language used should be improved throughout.
Author Response

(The authors gave the same response as above.)

Round 2
Reviewer 3 Report
Dear Authors,
The paper has significantly improved by these revisions. One point remains:
Line 786-801: This paragraph should be elaborated. It is unclear to me how nutrition and health knowledge education among children, adolescents, and their parents should be strengthened? What effective interventions/policies should be implemented to sustain behavior change? What is the role of Chinese government? Are there any studies discussed this point? I would suggest referring to these articles (e.g.,
Children (Basel). 2018 Jan 29;5(2):18.doi 10.3390/children5020018; Int J Environ Res Public Health. 2020, 17(22):8405).
Author Response
Dear Professor,
Special thanks to you for your good comments. We have studied comments carefully again and have made some correction which we hope meet with approval. The responds to your comments are as flowing:
Point 1: Line 786-801: This paragraph should be elaborated. It is unclear to me how nutrition and health knowledge education among children, adolescents, and their parents should be strengthened? What effective interventions/policies should be implemented to sustain behavior change? What is the role of Chinese government? Are there any studies discussed this point? I would suggest referring to these articles (e.g.,Children (Basel). 2018 Jan 29;5(2):18.doi 10.3390/children5020018; Int J Environ Res Public Health. 2020, 17(22):8405).
Response 1:
Thank you for your suggestion. The government should provide supportive policy options to reduce obesity in schools. These may include the development of policy options focused on nutrition/physical education and parental involvement in nutrition and physical activities[1]. The government should allow parents, teachers, and community health workers to participate in developing or revising childhood obesity control strategies, to bolster the comprehensiveness and strength of the district’s wellness policy language as well as accountability for these changes[2]. The strategy for capitalizing on opportunities to increase availability of schoolyard equipment should be promoted to enhance students’ physical activity. The health management department should work with after school program decision makers to ensure that the schedule includes 50% or more of time allocated to vigorous physical activity[2].
Health education for school students and their parents includes the following measures:(1) the use of posters in the classroom to teach children about healthy food and to encourage them to be physically active; (2) the provision of information skills such as recipe books or picture cards that increase children’s awareness and knowledge of healthy eating and physical activity; (3) letting the teachers have a good command theories of behaviour change to assist in maximizing children’s knowledge of activities during class time; (4) providing parents with information about healthy eating and physical activity through regular newsletters; (5) encouraging parents to be part of decision-making related to healthy school nutrition and the physical activity program.
The supplements are available in Line378-394.
We appreciate for your warm work earnestly. Thanks to you for your comments and suggestions.
[1]Escaron, A.L.; Martinez, C.; Lara, M.; Vega-Herrera, C.; Rios, D.; Lara, M.; Hochman, M. Program Evaluation of Environmental and Policy Approaches to Physical Activity Promotion in a Lower Income Latinx School District in Southeast Los Angeles. Int. J. Environ. Res. Public Health 2020, 17, 8405. https://doi.org/10.3390/ijerph17228405
[2]Alsharairi, N.A. Current Government Actions and Potential Policy Options for Reducing Obesity in Queensland Schools. Children 2018, 5, 18. https://doi.org/10.3390/children5020018
